# In-Hospital Antibiotic Use for COVID-19: Facts and Rationales Assessed through a Mixed-Methods Study

**DOI:** 10.3390/jcm11113194

**Published:** 2022-06-02

**Authors:** Laura Elena Stoichitoiu, Larisa Pinte, Alexandr Ceasovschih, Roxana Carmen Cernat, Nicoleta Dorina Vlad, Vlad Padureanu, Laurentiu Sorodoc, Adriana Hristea, Adrian Purcarea, Camelia Badea, Cristian Baicus

**Affiliations:** 1Faculty of Medicine, Carol Davila University of Medicine and Pharmacy, 050474 Bucharest, Romania; larisa.pinte@drd.umfcd.ro (L.P.); adriana.hristea@umfcd.ro (A.H.); cameliabadea72@yahoo.com (C.B.); cristian.baicus@umfcd.ro (C.B.); 2Department of Internal Medicine, Colentina Clinical Hospital, 020125 Bucharest, Romania; 3Clinical Research Unit Bucharest, Réseau d’Epidémiologie Clinique International Francophone, 020125 Bucharest, Romania; 4Internal Medicine, Grigore T. Popa University of Medicine and Pharmacy, 700111 Iasi, Romania; alexandr.ceasovschih@yahoo.com (A.C.); laurentiu.sorodoc@gmail.com (L.S.); 5Department of Internal Medicine, Clinical Emergency Hospital Sfantul Spiridon, 700115 Iasi, Romania; 6Faculty of Medicine, Ovidius University, 900527 Constanta, Romania; roxana.cernat@seanet.ro (R.C.C.); nicoleta.lalescu@yahoo.com (N.D.V.); 7Clinical Hospital of Infectious Diseases, 900178 Constanta, Romania; 8Faculty of Medicine, University of Medicine and Pharmacy Craiova, 200349 Craiova, Romania; vlad.padureanu@umfcv.ro; 9Craiova Emergency County Hospital, 200642 Craiova, Romania; 10National Institute of Infectious Diseases Prof. Dr. Matei Bals, 021125 Bucharest, Romania; 11Department of Internal Medicine, Sacele County Hospital, 505600 Brasov, Romania; adrian.purcarea@gmail.com

**Keywords:** COVID-19, SARS-CoV-2, antibiotics, antibacterial agents, mixed methods, qualitative, quantitative

## Abstract

It is well known that during the coronavirus disease 2019 (COVID-19) pandemic, antibiotics were overprescribed. However, less is known regarding the arguments that have led to this overuse. Our aim was to understand the factors associated with in-hospital antibiotic prescription for COVID-19, and the rationale behind it. We chose a convergent design for this mixed-methods study. Quantitative data was prospectively obtained from 533 adult patients admitted in six hospitals (services of internal medicine, infectious diseases and pneumology). Fifty-six percent of the patients received antibiotics. The qualitative data was obtained from interviewing 14 physicians active in the same departments in which the enrolled patients were hospitalized. Thematic analysis was used for the qualitative approach. Our study revealed that doctors based their decisions to prescribe antibiotics on a complex interplay of factors regarding the simultaneous appearance of consolidation on the chest computer tomography together with a worsening of clinical conditions suggestive of bacterial infection and/or an increase in inflammatory markers. Besides these features which might suggest bacterial co-/suprainfection, doctors also prescribed antibiotics in situations of uncertainty, in patients with severe disease, or with multiple associated comorbidities.

## 1. Introduction

Widespread antibiotic use leads over time to antimicrobial resistance, affecting all the countries, independent of their level of development. The Centers for Disease Control and Prevention reported that more than 2.8 million antibiotic-resistant infections occur in the U.S. every year, resulting in more than 35,000 deaths, but also in prolonged hospitalization, which represents a burden for the economy of any state [1]. Although the World Health Organization is constantly drawing attention towards the need for new antibacterial drugs to be developed, if advances in preventing the selection and the spreading of new resistant strains are not made and measures are not be implemented, then any new drug will have the same fate as the older ones [2,3]. One of the most common conditions in which antibacterial agents have been misused is in the management of patients (inpatients as well as outpatients) with various viral respiratory tract infections (RTIs). A previous study which enrolled 196 hospitalized patients with confirmed viral RTIs reported that 67% began antibiotic therapy, and 64% continued it after the confirmation of the viral infection, while 63% of the latter had normal chest-imaging findings [4]. At present, there is growing interest in the ways antibacterial agents were prescribed during the coronavirus disease 2019 (COVID-19) pandemic. It was shown that a minority of patients had a coinfection at admission (3.5 to 18.5%) or developed a secondary bacterial infection (3.8 to 14.3%), but more than 70% received antibiotics while hospitalized [5,6].

Currently, it is quite clear that antibiotics were overprescribed for both in- and outpatients with COVID-19 despite the relatively low rate of confirmed bacterial infections [5,6]. However, less is known about the rationale that has led to this antibiotic overuse. When reporting this high rate of antibiotic prescription to the enormous number of patients confirmed with COVID-19 since the pandemic emerged, it becomes obvious that we need to understand why the clinicians felt so tempted to give antibiotics. This information could be useful for further guidelines and antimicrobial stewardship programs that explore the aforementioned principles to be elaborated and implemented as support for doctors in guiding their decisions when managing future viral infections.

We aimed to understand the rationale behind antibiotic prescriptions in COVID-19, both by analyzing which factors proved to be associated with antibiotic treatment and by exploring the complex reasoning which ultimately served as grounds in this decision.

## 2. Materials and Methods

### Research Design

In our mixed-methods study we chose a convergent design to investigate antibiotic prescription during the COVID-19 pandemic. In a convergent design, quantitative and qualitative data are collected and analyzed separately, the final step consisting of mixing the results during the interpretation of the data in order to achieve a more comprehensive analysis [7]. Our aim was to assess if the results from the qualitative analysis are in agreement with the results from the quantitative approach. Moreover, we also wished to explore the potential disagreements that may arise given the fact that in clinical practice there is a complex cognitive process involving multiple pros and cons behind the decisions, which are almost impossible to be evaluated only through quantitative instruments. Therefore, our research question was the following: “What are the factors associated with in-hospital antibiotic prescriptions during the COVID-19 pandemic, and what are the doctors’ reasonings when deciding to administer antibacterial drugs?”.

## 3. Quantitative Approach

### 3.1. Study Design and Population

For this study, we used the same database as for the study of Pinte et al., which had the primary objective of assessing the impact of antibiotic treatment on the mortality of hospitalized patients with COVID-19 [8]. This was a prospective, multicenter, cohort study conducted in six institutions in Romania. We included adult patients confirmed with COVID-19, admitted between January 2021 and May 2021, who were divided into two groups according to the prescription of antibiotics (dependent variable, outcome). The study participants were enrolled from the departments of Internal Medicine, Pneumology, and Infectious Diseases.

The inclusion criteria for the study were patients 18 years of age or older confirmed with SARS-CoV-2 infection by a positive real-time polymerase chain reaction (RT-PCR) test or rapid antigen test. The exclusion criteria were patients initially admitted in the intensive care units (ICU), patients with end-stage kidney disease undergoing hemodialysis or peritoneal dialysis, and patients with hematologic malignancies. The treatment decision remained at the discretion of the attending physician. For the quantitative analysis we adhered to Strengthening the Reporting of Observational Studies in Epidemiology reporting guidelines [9].

### 3.2. Variables and Data Measurement

Patients were classified according to disease severity in agreement with the National Institutes of Health guidelines [10] as follows: mild (normal O_2_ saturation and normal chest X-ray), medium (radiological evidence of COVID-19 pneumonia), and severe disease (at least one of the following criteria: peripheral oxygen saturation (SpO_2_) ≤ 93% in ambient air, respiratory rate (RR) > 30/min, arterial oxygenation partial pressure to fractional inspired oxygen ratio (PaO_2_/FiO_2_ ratio) < 300, or lung infiltrates > 50% of lung parenchyma). During hospitalization, complete blood count, inflammation markers, and d-dimer values were obtained daily. We used the admission values for all patients and those prior to antibiotic administration (for patients who received antibiotics) and from the day with the greatest C-reactive protein (CRP) value (for patients who did not receive antibiotics). Patients who received oral vancomycin for *Clostridioides difficile* colitis were included in the non-antibiotic group.

### 3.3. Data Analysis

Demographic, clinical, biological, and imaging data of the enrolled patients were analyzed descriptively. Continuous and categorical variables were presented as median (min, max) and absolute numbers (percentage), respectively. The variables associated with *p* ≤ 0.10 in bivariate analysis with antibiotic prescription were introduced into a logistic regression model (forward stepwise selection) with prescription of antibiotics (yes/no) as the dependent variable. Statistical significance was set at *p* < 0.05. We analyzed the collected data using the Statistical Package for Social Sciences (SPSS version 20, IBM Corp., Armonk, NY, USA) and Microsoft Excel 2018 (Microsoft Corporation, Redmond, WA, USA).

## 4. Qualitative Approach

### 4.1. Methodology

For the qualitative analysis we used semi-structured, in-depth interviews with physicians from the same departments from where the patients were included. The Consolidated Criteria for Reporting Qualitative Research were used to report the methodology of the qualitative design [11]. Participants’ recruitment was directed via telephone, while the information and the consent forms were sent via e-mail. The interview was based on five questions which are presented in “Table 1”.

Additional questions were asked to ensure rich data collection. It was initially piloted on one person to establish if the questions we had designed would provide the needed data, but no modifications were made regarding the topic guide. The interviews were audio recorded and conducted face-to-face or over the phone according to the participants preference. All the interviews were transcribed verbatim by the interviewer, with the anonymization of the transcript. After the publication of the article, all the audio recordings will be destroyed.

### 4.2. Sample and Data Collection

Participants were not involved in the development of the research questions, study design, and recruitment process. Since the decision to prescribe or not antibiotics may vary with age and experience, we purposely selected respondents towards achieving maximum of variation in age. Volunteers received no remuneration.

### 4.3. Analysis

Given the research question, we conducted a primarily experiential form of thematic analysis using an inductive, data-driven approach, while focusing on both latent and semantic levels. We followed the stages described by Braun and Clarke, which consist of familiarization with the data, generating initial codes, actively searching for the themes, reviewing potential themes, defining and naming themes, and finally writing up the themes into a report [12]. We included in our report codes not only based on the saturation principle but also on the saliency analysis principle [13]. After familiarizing with the data, the interviewer (first author) generated the codes and presented them to the last author, who was also the supervisor of the study; together we matched the codes into themes in three meetings. The report was then written and sent to three randomly selected participants to perform member checking. We achieved data saturation after 14 interviews.

### 4.4. Ethical Considerations

This study was performed in line with the principles of the Declaration of Helsinki and accepted by the Ethics Committee of the involved medical centers (32/08.12.2020). Patients signed an written informed consent during their hospital admissions, while the doctors who were enrolled in the qualitative analysis signed the informed consent before the interviews.

## 5. Results

### 5.1. Quantitative Approach

A total of 553 patients were included in the study of which 311 (56.2%) received antibiotics. The median time from admission until antibiotics prescription was 0 (min 0, max 24) days. Patients’ characteristics at admission and at the moment of antibiotic initiation (for the patients who received antibiotics)/the day with the highest CRP value (for the patients who did not receive antibiotics), together with routine tests results, and the treatment they received for COVID-19 are presented in Table 2.

In our study, the variables associated with antibiotic prescription were older age, higher Charlson Comorbidity Index, COVID-19 severity, the presence of pulmonary infiltrates and pulmonary consolidation on CT scan, higher procalcitonin, WBC and neutrophils levels, but not higher inflammation markers values (CRP, ferritin). However, after adjusting for the pulmonary consolidation, the pulmonary infiltrates were no longer associated with antibiotic prescription. In Table 3, after adjusting for the factors related to antibiotic administration in bivariate analysis, only the presence of pulmonary consolidation, a higher Charlson Comorbidity Index, and higher neutrophil count remained independent factors associated with antibiotic prescription. (Table 3). This regression model predicted antibiotic prescription with an AUROC (95% CI) of 0.791 (0.751, 0.830).

### 5.2. Qualitative Approach

For the qualitative part of the study, we interviewed 14 physicians. The ages ranged from 29 to 57 years old; further characteristics of the doctors whom we interviewed are presented in Table 4. 

After we analyzed and coded the transcripts, we identified two themes which are defined in Table 5: “Times have changed” and “Justifying antibiotic prescription” with the second theme having two subsequent subthemes. 

## 6. Times Have Changed

Before SARS-CoV-2 emerged, elevated values of inflammation markers and/or procalcitonin were linked to bacterial infection. Nowadays, when almost all patients admitted to the hospital have high levels of CRP and/or procalcitonin, together with clinical signs of pulmonary distress, doctors are tempted to associate this with a concomitant bacterial infection, thinking that maybe SARS-CoV-2 alone cannot produce biological abnormalities of such a magnitude. 

“We were used to prescribe antibiotics based on criteria regarding inflammation: CRP, ESR, and sometimes procalcitonin, and often leukocytosis, neutrophilia, and of course fever and chills. Now, due to the high prevalence of this viral infection which is associated with a marked inflammation, we tend to directly treat this inflammation, and we give much more antibiotics based only on CRP […] or maybe we directly treat an elevated procalcitonin”(*Physician 1*)

However, as time went by, and more data emerged, most doctors realized that they could no longer approach the diagnosis of a bacterial infection based on biological inflammation, which is difficult to be used as a reason for “here is a bacterial infection, we have to give antibiotics.” (*Physician 2*)

Besides the inflammation, many doctors felt that they could no longer use other laboratory markers of bacterial infection, such as leukocytosis and neutrophilia, nor clinical markers, such as fever and chills.

“Those patients…they don’t develop fever, many of them… or, what kind of sepsis is this, if you don’t have fever, you don’t have leukocytosis, you only have a CRP which is rising, and procalcitonin…if procalcitonin is good, at least you are somehow more comfortable […] this is one question that I keep asking myself… either these patients with COVID-19 do not develop leukocytosis, or those patients who did not develop leukocytosis did not have a bacterial infection and we prescribed them antibiotics for nothing”(*Physician 7*)

However, in the conundrum of inflammatory marker cut-offs, procalcitonin was the one that lead to the most divided opinions, with some doctors guiding the prescriptions of antibiotics based on their value and previous thresholds, while others put it quite in the same place with the CRP levels, considering that higher cut-off values would be more appropriate, even though, before COVID-19, procalcitonin represented a strong argument in favor of bacterial infections, as it is shown below.

“Once again, now we are resisting even when we see a procalcitonin of 1 or 2, and before, when we saw this level of procalcitonin, we were saying that it is clearly sepsis”(*Physician 7*)

“Elevated procalcitonin. Everything that was even at the upper level of normal, I think that this was the point when I prescribed. If procalcitonin was somehow elevated, then I think that I jumped and I prescribed antibiotics”.(*Physician 8*)

Left with few rapid strong arguments to diagnose a bacterial infection in an incipient phase and given the fact that previous cut-offs were no longer usable, doctors felt out of their comfort zone. Moreover, due to the enormous number of cases, some doctors were forced into treating patients with severe COVID-19, even though they were not used to treating this kind of pathology or patients with such severe respiratory distress. As consequence, they sometimes overtreated in order to feel the comfort of knowing that everything was done, while the need to cover a possible bacterial coinfection was frequently the main source of discomfort. Even though they had in mind the risks of developing an overwhelming antibiotic resistance in time, they considered that the risk of not treating a possible bacterial infection which would have explained the patients’ symptoms, especially when you deal with a patient with a rapidly declining status, would have been much worse. Therefore, they often decided to do what seemed to be the best at that point, rather than keep worrying about complications which would appear after a long period of time.

“I am always comparing with how I would feel if I were to work in a ward dealing with acute coronary syndromes…probably I would feel the same temptation…to administer any kind of medication in order to alleviate the symptoms that I am not used with, and I think that this is what everyone would do”(*Physician 10*)

“When you are in a dilemma, you give what you consider that you should give, without any reproach, because you are in a dilemma, which means that you are outside of your comfort and expertise area, and until you build in, you have to react in a way that it is not mandatory to be 100% cortical, because you don’t have the experience”(*Physician 5*)

## 7. Justifying Antibiotic Prescriptions

### 7.1. Clear Indications

When asked about the clear reasons that are decisive in favor of prescribing antibiotics, besides a positive culture, doctors exposed intricate cognitive processes involving clinical symptoms, biological markers, and imagistic abnormalities, which go beyond a unique sine qua non factor. Therefore, most of them considered that the simultaneous appearance of consolidation on the chest computer tomography together with a shift in the patients’ clinical status suggestive of bacterial infection, such as productive cough, chest pain, oxygen desaturation, or simply an alteration of the clinical status, or/and an elevation of the inflammation, was suggestive of bacterial infection; therefore, in these situations, physicians felt entitled to initiate antibiotic therapy along with active searching of the pathogen agent.

“I would give antibiotics with all my heart when there are clinical elements that suggest bacterial coinfection […] productive cough with purulent sputum from a clinical point of view… and from an imagistic point of view, a pattern of alveolar consolidation, in the detriment of interstitial abnormalities”(*Physician 2*)

“An aggravation of the respiratory function, fever, usually when you don’t expect for such abnormalities to appear, which means after many days since the symptoms of COVID-19 started, and all these things, of course, in the context of an elevation of the inflammation, whether it is accompanied or not by an elevated procalcitonin level”(*Physician 13*)

As it is illustrated above, no physician based their decision to prescribe antibiotics solely on one factor. They had to have more determinants, usually from the main three possible sources (clinical, biological, imagistic) to decide to administer antibiotics. Besides this mixture of determinants, another important aspect in the decision of prescribing or not prescribing antibiotics consisted in the timing of the moment when there was a shift in the clinical/paraclinical status of the patients. Thus, as it is shown below, if the abnormalities appeared soon after the onset of COVID-19 symptoms, the doctors considered that the deterioration was due to the aggravation of SARS-CoV-2 infection, rather than to bacterial overgrowth.

“It mattered in taking the decision, when the patient came to us, because if the patients were hospitalized in the first days of the symptoms’ onset, then…… uuummm in the first 7–8 days, when the clinical picture is the most obvious, then I would wait to pass over this period. If the patient presented to us in the eighth or tenth day of the disease, or later, than I did not wait, because the chance for SARS-CoV-2 infection to be the explanation would be very low”.(*Physician 3*)

### 7.2. When More Is Better

Even though physicians had in mind which were the clear indications for antibiotics prescription, many grey areas arose in practice when the feeling that more is better was legitimate in their perception, and consequently, they acted as such. In many cases, the balance between a bacterial infection versus COVID-19 aggravation represented the hardest decision to be made, considering that COVID-19 aggravation and the inflammatory storm may appear later in the disease evolution, which in many cases overlapped with a prolonged hospitalization, while the latter itself could have been a factor for bacterial coinfection or a hospital-acquired infection. Moreover, many hospitalized patients were frail, with multiple comorbidities, and received immunomodulators as a treatment for COVID-19, and for them, not treating a bacterial infection in an incipient phase could have been fatal.

“The problem with these patients is that they come to the hospital for COVID-19, for a while they are well, and after that the CRP levels increase, and you always ask yourself… eventually with a degradation of the clinical status… and then the question is: is it the second phase of the disease, the cytokine storm, the hyperimmune phase, or is it a coinfection?”(*Physician 7*)

Therefore, the more is better principle arose in three settings which frequently overlapped: when the clinical status of the patient was very deteriorated, no matter the presence or the absence of previous comorbidities; when the patient had been aggressively treated with immunomodulators due to the severity of the COVID-19 disease; and when the patients were frail with multiple comorbidities, including diseases associated with immunosuppression.

“The patient who is very severe and very fragile…sometimes you do not have time to wait… you have to give him antibiotic because you do not have much to lose at this point, and you have to save him no matter what…and if… if the antibiotic may be that saving element, and it must be prescribed early… I mean, you should not hesitate, you do not have time to hesitate”(*Physician 13*)

“For example, if I want to treat a patient with immunomodulators, even if he has a colonization of the urinary tract, even if he has no complaints […] if I have signs of an infection, a subclinical one, I would probably treat it, in a minimal fashion, five days a cystitis with the “easiest” or the most targeted antibiotic”(*Physician 5*)

## 8. Discussion

Our study revealed that doctors based their decision to prescribe antibiotics on a complex interplay of factors regarding the simultaneous appearance of consolidation on the chest-computed tomography together with a shift in the patients’ clinical status suggestive of bacterial infection and/or an increase in inflammatory markers. The timing when the symptoms appeared, together with the Charlson Comorbidity Index, and the severity of the disease also played an important role in their choice. Besides these clear indications, doctors also decided to prescribe antibiotics in situations of uncertainty, when they considered that the “more is better” principle is appliable. 

One of the main problems encountered during the COVID-19 pandemic regarding antibiotics prescription revolved around the fact that the clinical and paraclinical picture of the patients changed. Most of the patients had significant inflammation, while in the doctors’ opinions, few of them (with confirmed bacterial infection or in sepsis) had associated markers of “traditional” bacterial infection, such as fever, leukocytosis, neutrophilia, or productive cough. Fever and productive cough did not correlate with antibiotic prescription in our study as opposed to the findings of Estrada et al. [14], which may have happened because these clinical symptoms appeared less often in practice (7.6% of the patients had productive cough, 17.2% had fever) than in previous times—inability to expectorate tracheobronchial secretions and fever blunted by corticosteroids use. Neutrophilia was strongly associated with antibiotic administration, in agreement with other results [14,15,16]. 

At first, doctors felt tempted to prescribe antibiotics based on elevated markers of inflammation, reminiscent from previous times when a high CRP value was frequently associated in clinical practice with a bacterial infection. As time went by, they realized that given the cytokine storm associated with the SARS-CoV-2 infection, this usual marker was no longer useful. Analyzing the qualitative data, only procalcitonin remained a useful argument for associated bacterial infection, its cut-off value being however debatable, with some of the doctors considering that, in COVID-19, higher diagnostic values would be more appropriate. This was further confirmed in the quantitative analysis, where high CRP values were not associated with antibiotic prescription, as opposed to procalcitonin, which showed a strong association. This may be since even though every doctor had different thresholds for bacterial infections when referring to procalcitonin, they used it to guide their prescriptions, while most of them considered elevated CRP values nonspecific, being also associated with COVID-19 aggravation. However, previous studies reported an association not only between procalcitonin but also between higher CRP values and antibiotic prescription, which may be explained by the fact that the patients were enrolled early in the pandemic when less was known about the cytokine storm associated with COVID-19 [14,15,16,17]. Regarding procalcitonin’s utility in diagnosing associated bacterial infections in patients with COVID-19, a recent study showed that a value <0.25 ng/mL has a negative predictive value of over 95% for bacteremia or bacterial pneumonia, but higher procalcitonin levels also predict COVID-19 severity in hospitalized patients [18]. Moreover, a meta-analysis also showed that elevated procalcitonin levels were associated with a nearly five-fold higher risk of developing a severe form of COVID-19, but this data needs to be cautiously interpreted since no analysis according to the presence or absence of associated bacterial infections was done [19]. Therefore, in agreement with doctors’ opinions exposed in the qualitative analysis, procalcitonin is useful in guiding antibiotic treatment—low levels of procalcitonin shows that associated bacterial infections are unlikely, but high levels of procalcitonin are not diagnostic for bacterial coinfections since they may be due to COVID-19 related immune dysfunction.

Regarding the criteria for antibiotic prescription, doctors did not base their decisions solely on one reason, but rather on an interconnection of factors of which alveolar consolidation on computed tomography examination was highly predictive for antibiotic prescription. Previous qualitative studies regarding antibacterial drug use during COVID-19 were mostly developed in primary care settings where the clinical scenario along with the disease severity were completely different; therefore, we could not compare our findings regarding the complex rationale behind antibiotic prescription. However, a previous qualitative study published before the emergence of COVID-19 showed that physicians were likely to base their decisions to administer antimicrobial drugs based solely on clinical grounds, which were no longer appliable given the fact that the patient–physician interactions were severely shortened due to the risk of SARS-CoV-2 transmission [20]; this idea was also presented by Borek et al. in a qualitative study which involved general practitioners from England [21]. Regarding the presence of lung consolidation, in the Estrada et al. study [14], not only alveolar infiltrates but also interstitial infiltrates were linked to antibiotic prescriptions, while the presence of bilateral interstitial infiltrates were strongly associated with what was considered inappropriate antibiotic use in the study of Calderón-Parra et al. [16]. In our study, pulmonary infiltrates (without consolidation) were not a driver for antibiotic therapy as 90% of the patients who did not receive antibiotics had such infiltrates. 

Besides clear criteria for antibiotic therapy, in qualitative analysis, all the clinicians considered that in some cases, more is better when it comes to antibiotic administration; most often, these cases were represented by an important deterioration of the patients’ clinical status, iatrogenic immunosuppression (through immunomodulators for COVID-19), and patients’ frailty (multiple comorbidities associated). Antibiotics overuse was previously reported to be associated with clinical uncertainty, when prescribing them was perceived to be a safer option during COVID-19 times, but also before the pandemic state [21,22]. However, the prescription of neither tocilizumab nor anakinra was associated with antibiotic therapy in quantitative analysis, probably due to a selection bias—doctors prescribed immunomodulators in patients who were unlikely to have an associated bacterial infection or in patients in whom such an infection was excluded. Therefore, immunomodulators administered on their own were not an important driver for antibiotic prescription. Although in a retrospective cohort study which took place early in the pandemic was launched the idea that antibiotic prophylaxis in tocilizumab use would be beneficial [23], we did not find previous studies to assess the association between the use of potent immunomodulators and antibiotic use.

Both the severity of the disease and the Charlson Comorbidity Index were associated with antibiotic prescription, which was further confirmed in the qualitative approach; physicians considered that in some situations of uncertainty—severe deterioration of the patients’ clinical status, treatment with immunomodulators, and the presence of multiple comorbidities—antibiotic prescriptions were justifiable, in order to avoid an unfavorable outcome. 

The strength of this study resides in its mixed-methods design with the quantitative part being approached through a prospective, multicenter study. Given the fact that our aim was not to assess the degree of in-hospital implemented antimicrobial stewardship, we did not perform analyses regarding the appropriate vs. inappropriate antibiotic administration, but we evaluated which factors were the determinants in deciding whether to prescribe or not antibacterial drugs. One of the limitations of the study resides in the fact that we involved patients (quantitative study) and physicians (qualitative study) from only three specialties: internal medicine, pneumology, and infectious diseases, having in mind the fact that other specialties may have had other practices regarding antibiotic prescription. In our cohort, the number of patients enrolled from the infectious diseases departments was relatively small, and therefore, we could not perform a subgroup analysis.

Overall, we identified that physicians chose to prescribe antibiotics also in situations of uncertainty (in patients with a severe form of the disease or with multiple associated comorbidities), while in their opinion, the simultaneous appearance of abnormalities in the patients’ clinical status, biological markers, and pulmonary-computed tomography represented a clear indication for in-hospital antibiotic use during the COVID-19 pandemic. Given the fact that antibiotic overuse in viral respiratory tract infections represents a common problem and still poses a challenge, these kinds of studies are important to be conducted so that the key arguments in doctors’ views are identified and understood, in order to improve further antibiotic prescriptions in viral infections through targeted antimicrobial stewardship programs. Therefore, periodic in-hospital trainings regarding the peculiarities of patients with viral infections with and without associated bacterial infections (imagistic abnormalities, the role of inflammatory markers, rate of bacterial coinfections, criteria of certainty when it comes to antibacterial drugs administration) should be done to reduce treating out of the fear of missing infections (FOMI) and its inherent consequences. 

## Figures and Tables

**Table 1 jcm-11-03194-t001:** Interview topic guide.

In Your Opinion, How Often Do You Prescribe Antibiotics to COVID-19 Patients?
1. Which arguments/settings represent in your opinion a clear indication for antibiotic prescription in COVID-19 patients?
2. What are the arguments, or in which situations do you prescribe antibiotics in COVID-19 patients without having a certainty regarding the presence of an associated bacterial infection?
3. How do you differentiate between colonization and infection?
4. Do you consider your antibiotic prescription practices changed during the pandemic? How about when comparing the emergence of the pandemic with the actual moment when we have some experience in treating COVID-19 patients?

**Table 2 jcm-11-03194-t002:** The distribution of the variables according to antibiotic prescription.

Variable	Antibiotics N = 311	Non-Antibiotics N = 242	AUROC (95% CI)	*p*-Value
Gender, male, N (%)	159 (51.1)	124 (51.2)		1
Age, median (min, max)	70 (32, 94)	65 (18, 92)	0.599 (0.551, 0.647)	<0.001
Charlson Comorbidity Index, median (min, max)	4 (0, 12)	3 (0, 12)	0.668 (0.622, 0.713)	<0.001
Disease severity, N (%)	311 (56.2)	242 (43.8)		<0.001
Mild	19 (6.1)	25 (10.3)		
Moderate	148 (47.6)	149 (61.6)		
Severe	144 (46.3)	68 (28.1)		
Pulmonary infiltrates, N (%)	298 (95.8)	217 (89.7)		0.006
Corticosteroid treatment, N (%)	237 (76.2%)	194 (80.2)		0.301
Tocilizumab, N (%)	13 (6.8%)	13 (5.4%)		0.594
Anakinra, N (%)	48 (15.4%)	41 (16.9)		0.643
Fever *, N (%)	48 (15.4)	44 (18.2)		0.421
Productive cough, N (%)	28 (9)	14 (5.8)		0.196
Symptoms of UTI, N (%)	5 (1.6)	2 (0.8)		0.476
Pulmonary consolidation on CT, N (%)	173 (55.6)	30 (12.4)		<0.001
SpO_2_ at ATB p, median (min, max)	93 (53, 99)	93 (56, 99)		0.309
CRP *, median (min, max)	66.2 (0.2, 390.6)	61.5 (0.26, 312.2)	0.513 (0.462, 0.564)	0.614
Procalcitonin *, median (min, max)	0.15 (0.02, 24.8)	0.08 (0.02, 5)	0.671 (0.610, 0.732)	<0.001
Ferritin, median (min, max)	615.2 (58, 5887)	496 (6, 3993)	0.548 (0.496, 0.600)	0.089
WBC *, median (min, max)	8810 (1060, 29,760)	7100 (1205, 25,100)	0.634 (0.585, 0.683)	<0.001
Neutrophils *, median (min, max)	7160 (650, 26,400)	5240 (660, 20,000)	0.638 (0.589, 0.686)	<0.001
Lymphocytes *, median (min, max)	1005 (150, 5930)	1065 (260, 3500)	0.493 (0.441, 0.544)	0.788

* At the moment of antibiotic initiation (for the patients who received antibiotics)/the day with the highest CRP value (for the patients who did not receive antibiotics). Abbreviations: ATB—antibiotic, UTI—urinary tract infection, CT—computed tomography, SpO_2_—oxygen saturation level, CRP—C-reactive protein, WBC—white blood count.

**Table 3 jcm-11-03194-t003:** Factors associated with antibiotic prescription (logistic regression).

Variables	B	OR	95% CI for OR		*p*
			Upper	Lower	
Charlson Comorbidity Index	0.177	1.193	1.071	1.330	0.001
Pulmonary consolidation	1.907	6.732	3.323	13.641	<0.001
Neutrophil count	0	1.000	1	1	0.001

**Table 4 jcm-11-03194-t004:** Participants’ characteristics.

Participants		Numbers
Age	<30	2
	30–50	7
	>50	5
Gender	F	6
	M	8
Function	Senior Physician	12
	Resident Physician	2
Specialty	Internal Medicine	8
	Pneumology	1
	Infectious Diseases	5

**Table 5 jcm-11-03194-t005:** Overview of themes.

Themes Titles	Themes Definitions	Subthemes
Times have changed	This theme explores the difficulties perceived by physicians in the management of patients with COVID-19 due to the fact that the whole pattern of the patients changed from a clinical, as well as from a laboratory point of view when previous cut-offs of inflammatory markers were, in their opinion, no longer worthy to count on.	
Justifying antibiotic prescriptions	This theme explores the reasons why doctors prescribed antibiotics by approaching the clear indications for this practice, in addition to the equivocal determinants, to achieve a larger frame.	Clear indications
		When more is better

## Data Availability

Supporting data for the quantitative approach is available upon request. Given the fact that the supporting data for the qualitative approach includes sensitive information which may lead to the identification of the physicians who agreed to participate in the study, it may not be shared publicly; therefore, the data is available upon request from the Colentina Hospital Ethics Committee of Research for researchers who meet the criteria to access confidential data and only after the complete removal of the sensitive information from the transcripts. (Colentina Hospital Ethics Committee of Research at Colentina Clinical Hospital, Soseaua Stefan cel Mare 19–21, sector 2, 020125, Bucharest, Romania—President of the Colentina Hospital Ethics Committee of Research: Gheorghe Andrei Dan, andrei.dan@gadan.ro).

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
