# Peer review of "In-Hospital Antibiotic Use for COVID-19: Facts and Rationales Assessed through a Mixed-Methods Study"

_jcm, 2022, doi:10.3390/jcm11113194_

Round 1
Reviewer 1 Report
ear authors, many thanks for the opportunty to read your great paper
Introduction is concise and well written,
methods and results are clear
My only my concern regards the discussion: I think the topic of immunosupression the use of tocilizumab or steroids could have influenced the choice of antibiotics use: by the way, some studies demonstrated tha the infection rate was similar (https://doi.org/10.3390/jpm11111234)
I should only suggest to improve this part of discussion: minor revision
Author Response
Thank you for the effort and time spent, we hope we managed to make the article better.
Dear authors, many thanks for the opportunty to read your great paper
Introduction is concise and well written,
methods and results are clear
My only my concern regards the discussion: I think the topic of immunosupression the use of tocilizumab or steroids could have influenced the choice of antibiotics use: by the way, some studies demonstrated tha the infection rate was similar (https://doi.org/10.3390/jpm11111234)
We emphasised in the Discussion section that in the qualitative analysis it was pointed out that the use of immunosupression may be a factor for antibiotic administration, but in the quantitative analysis neither tocilizumab nor anakinra use was associated with antibacterial drug prescription. As we mentioned in the main text, we did not find any studies to asses the association between the use of these immunomodulators and antibiotic use - in the research paper you suggested, even though the infection rate was similar, patients with known active infection were excluded, and there are no information regarding the rate of antibiotic prescription.
I should only suggest to improve this part of discussion: minor revision

Reviewer 2 Report
Stoichitoiu et al. present a study on the self-reported reasons physicians give for their prescription of antibiotics in COVID-19 patients in 6 hospitals in their country.
Even though the quantitative study presentation is quite straightforward, the qualitative part requires additional work in the presentation. 5.2 and 6.1 parts are way to extensive and I believe it would be preferable if there are tables with main themes and some indicative quotes next to them. Not all of the quotes need to be in in the manuscript and if the authors feel that they are needed they can be added as supplemental material.
I have never come across another qualitative study that had only 2 themes identified. The two presented are not making much sense as themes but perhaps more as groups of themes for which again I would consider "reasons for prescribing" (elevated inflamation markers, unknown condition, fragility of patient) "reasons for not prescribing" or "self-protection" (dont know what I am dealing with - I prescribe) / "peer-mimicking" - that's what the others do.
It would be interesting to see a comparison of ID physicians vs the rest.
Overall there were only 12 physician asked. were all 6 centers represented equally, both in the quantitative and qualitative study? Otherwise we might be looking quantitative results from one center that prescribed more antibiotics than the others and qualitative results from the physicians of a totally different center. How were patients and interviewees selected in order to ensure representability and comparability of the two arms of this study?
How was the number of 14 physicians selected? why not more or why not less? Are residents usually prescribing or are they following senior physicians' orders? If so then their opinion since they are not the actual prescribers are not very relevant.
Perhaps one paragraph in discussion on how would these results be used in a stewardship effort would make the importance of this study more evident.
L160 - what is member checking?
Author Response
We thank the reviewer for the effort and time spent, and also for his insightful comments.
Stoichitoiu et al. present a study on the self-reported reasons physicians give for their prescription of antibiotics in COVID-19 patients in 6 hospitals in their country.
Even though the quantitative study presentation is quite straightforward, the qualitative part requires additional work in the presentation. 5.2 and 6.1 parts are way to extensive and I believe it would be preferable if there are tables with main themes and some indicative quotes next to them. Not all of the quotes need to be in in the manuscript and if the authors feel that they are needed they can be added as supplemental material.
We do not have neither in our article, nor in the results section 5.2 or 6.1 parts; however, after rereading the results section we agree with the fact that in some cases, the quotes were too extensive; therefore, we removed some of them as you suggested. We took into consideration adding some quotes in Table 6, but in the end, after adding the quotes, the table was too stiff with information and difficult to follow; therefore, we decided too keep the original format of the table.
I have never come across another qualitative study that had only 2 themes identified. The two presented are not making much sense as themes but perhaps more as groups of themes for which again I would consider "reasons for prescribing" (elevated inflamation markers, unknown condition, fragility of patient) "reasons for not prescribing" or "self-protection" (dont know what I am dealing with - I prescribe) / "peer-mimicking" - that's what the others do.
Given the fact that the design of the study is mixed-methods, our aim was not to show in the results section an extensive analysis of all the data encountered in the interviews. Instead, our aim was to compare the results from the quantitative analysis to the accordant information which emerged from the qualitative analysis, and this is the reasoning for which we only presented two themes. Moreover, we consider that if we had presented more themes, the qualitative analysis would have overcome the information presented in the quantitative analysis and consequently, the article would have been difficult to read and to follow. Regarding the themes, we considered important to present the framework in which physicians carried out their work (first theme), in order to gain a better understanding of their state of mind and consequently to understand the factors that led to their decisions regarding antibiotic therapy. The second theme, as you mentioned, addresses the reasons for which the physicians decided to prescribe antibiotics, discussing both the factors of certainty, and also the ones that led to antibiotic administration due to the fear of missing infection. We did not consider suitable to approach the situations in which they decided to not prescribe antibacterial drugs, because these are easily identified through logical deduction, and therefore the information would have become redundant. Concerning „self-protection”, we did not find this as a driver for antibiotic prescription in the case of fear of missing infection, but rather the aim of patient protection.
It would be interesting to see a comparison of ID physicians vs the rest.
Even though we were expecting to, we did not encounter important differences in qualitative analysis between specialists in infectious diseases and the other doctors; therefore, we did not perform subgroup analysis.
Overall there were only 12 physician asked. were all 6 centers represented equally, both in the quantitative and qualitative study? Otherwise we might be looking quantitative results from one center that prescribed more antibiotics than the others and qualitative results from the physicians of a totally different center. How were patients and interviewees selected in order to ensure representability and comparability of the two arms of this study?
The representativeness of the patients was fully ensured by the fact that ALL the patients admitted in the 6 participant centers during the 3rd wave were included, given that they fulfilled the inclusion criteria. Regarding the physicians enrolled in the qualitative analysis, we enrolled an equal number from each center. However, it must be stressed the fact that, contrary to the quantitative research, where a probability sample is needed to enable the statistical inference to a population, in the qualitative research it is used a purposeful sample, therefore we selected the physicians we considered that could best inform about the drivers for prescribing antibiotics. With regard to the „comparability between the two arms”, in our opinion, this can not be completely achieved, due to the design of the study itself, mixed-methods, which implies two different approaches (quantitative and qualitative) with two completely different types of participants (patients and physicians).
How was the number of 14 physicians selected? why not more or why not less? Are residents usually prescribing or are they following senior physicians' orders? If so then their opinion since they are not the actual prescribers are not very relevant.
The number of participants was determined according to the saturation principle, one of the main principles in qualitative analysis; according to it, the number of participants, in order to ensure rich data collection, is not predetermined as it is in quantitative analysis, but it rather „emerges” along with the interviews, enrolling new participants being not necesary when the information generated through the interviews becomes redundant (saturation). Therefore, in our study, saturation was reached after 14 interviews, and after this point, we did not enroll any new physicians. The residents enrolled in the qualitative analysis are senior residents (last year of preparation), who have full autonomy, and who prescribed antibiotics according to their own judgement and principles, while their supervisor was only observing their decisions and intervened in selected cases to modify their prescriptions.
Perhaps one paragraph in discussion on how would these results be used in a stewardship effort would make the importance of this study more evident.
We added a paragrapgh in the Discussion section.
L160 - what is member checking?
Member checking or member validation it is regarded, according to Lincoln and Guba, as a qualitative version of reliability, and it involves sending an written draft to random participants in the qualitative study in order to ensure if there is a fit between participants’ views and the authors interpretation.

Round 2
Reviewer 2 Report
no comments
Author Response
Thank you.